# A Rare Case of Blastic Plasmacytoid Dendritic Cell Neoplasm in a Child Mimicking Lymphoma/Leukemia Cutis

Phanitchanat Phusuphitchayanan [1,*], Voraphol Vejjabhinanta [2], Chayamon Takpradit [3], Poonnawis Sudtikoonaseth [4], Manasmon Chairatchaneeboon [5], Thamonpan Kiatvichukul [3] and Sanya Sukpanichnant [6]

1. Institute of Dermatology, Department of Medical Services, Ministry of Public Health, Bangkok 10700, Thailand
2. Dermatologic Surgery and Laser Division, Institute of Dermatology, Department of Medical Services, Ministry of Public Health, Bangkok 10400, Thailand
3. Department of Pediatrics, Faculty of Medicine Siriraj Hospital, Mahidol University, Bangkok 10700, Thailand
4. Dermatopathology Division, Institute of Dermatology, Department of Medical Services, Ministry of Public Health, Bangkok 10400, Thailand
5. Department of Dermatology, Faculty of Medicine Siriraj Hospital, Mahidol University, Bangkok 10700, Thailand
6. Department of Pathology, Faculty of Medicine Siriraj Hospital, Mahidol University, Bangkok 10700, Thailand
* Correspondence: p.phusuphitchayanan@gmail.com; Tel.: +66-94-969-6245

**Abstract:** Blastic plasmacytoid dendritic cell neoplasm (BPDCN) is a rare tumor that affects elderly individuals and presents a poor prognosis. Skin is the most common site of involvement, accounting for 89% of the cases. Extracutaneous organs, especially bone marrow, lymph nodes, and peripheral blood, can be involved at the time of diagnosis. We report a case of BPDCN in a child, presenting with a cutaneous lesion mimicking lymphoma or leukemia cutis. The histologic findings revealed a dense diffuse infiltration by monomorphic agranular medium-sized blast cells with sparing of the grenz zone, whose first immunophenotypic profile raised the possibility of diagnosing B lymphoblastic lymphoma or leukemia. However, the absence of CD10 expression and strongly positive expression for CD4, CD56, CD45RA, and the plasmacytoid dendritic cell-associated antigens, including CD123, supported the definite diagnosis of BPDCN. The patient responded well to a systemic combination chemotherapy regimen, modified from the Associazione Italiana Ematologia Oncologia Pediatrica (AIEOP) protocol for anaplastic large cell lymphoma (ALCL), that differed from the established recommendation using ALL protocol. Owing to the patient's excellent treatment outcome, this regimen could represent an effective alternative regimen for BPDCN in children.

**Keywords:** blastic plasmacytoid dendritic cell neoplasm; plasmacytoid dendritic cell; lymphoma cutis; leukemia cutis

## 1. Introduction

Blastic plasmacytoid dendritic cell neoplasm (BPDCN) is a well-defined but rare clinically aggressive hematological malignancy derived from plasmacytoid dendritic cells (PDC), originating from hematopoietic stem cells [1]. An international survey of 398 adult patients with BPDCN collected from 75 centers in five countries showed a male-to-female ratio of 3:1 and the median age of 67 years (range 18–96 years). The skin was the most common site of involvement at diagnosis (89%), followed by bone marrow (62%), lymph nodes (39%), and peripheral blood (15%). Isolated skin lesions were found in 30% of cases and cutaneous dissemination without extracutaneous involvement in 50% of cases [2].

BPDCNs are extremely rare in children. Between 1996 and 2009, there have only been nine cases of pediatric BPDCN at the US National Cancer Institute (NCI) and 20 additional pediatric cases have been previously published [3]. Seven of the 29 cases lacked cutaneous

disease at presentation and had a favorable outcome. Pediatric patients showed a less aggressive clinical course than adults. Treatment with chemotherapy is seemingly effective, and stem cell transplantation could be reserved for relapsed patients.

BPDCN has the same morphological features at any age, characterized by diffuse monomorphic small to medium or medium-sized mononuclear cells with blastic/blastoid nuclei resembling leukemic infiltration. Fine nuclear chromatin and one to several small nucleoli with irregular nuclear contour are the key features [1,3,4]. The scant and agranular cytoplasm appears on Giemsa or Wright staining. However, cells with plasmacytoid feature have never been described in the neoplasm despite the presence of "plasmacytoid" in the name of BPDCN. In cutaneous involvement, extensive diffuse dermal infiltration by neoplastic cells with extension to the subcutaneous fat is common; however, the epidermis and grenz zone, the uppermost rim of the superficial dermis, are spared [1,3,4].

Immunophenotypic profiles are typically positive for CD4, CD56, CD43, CD45RA, and PDC-associated antigens, including CD123, CD303, TCL1a, CD2AP, SPIB, MX1, and TCF4. However, neoplastic cells can show varied positivity for BCL2, BCL6, CD2, CD5, CD7, CD33, CD34 (by flow cytometry), CD38, CD68, CD79a, CD117, IRF4/MUM1, S-100, and TdT, resulting in misdiagnosis of BPDCN as lymphoma, leukemia, or other round cell neoplasms [1].

We report a case of BPDCN in a child presenting with a cutaneous lesion resembling lymphoma or leukemia cutis. The patient responded positively to systemic combination chemotherapy.

## 2. Case Report

A nine-year-old boy presented with a seven-month-old spontaneous enlarging erythematous subcutaneous tumor on his left forearm. Within one month, the lump gradually enlarged and ulcerated (Figure 1). The patient had no B symptoms. He had received oral doxycycline and clarithromycin for 10 days prior to consultation with no improvement. No other complaint or family history was obtained. Physical examination revealed a solitary, well-defined, large, erythematous, rubbery, fixed, non-tender mass with crusting on the left forearm, measuring 8 cm in diameter, and a non-tender left axillary lymph node, measuring 2 cm in diameter. Other physical findings were within normal limits. Differential diagnoses included chronic infection, lymphoproliferative disorder, lymphoma/leukemia cutis, and other non-hematologic tumors.

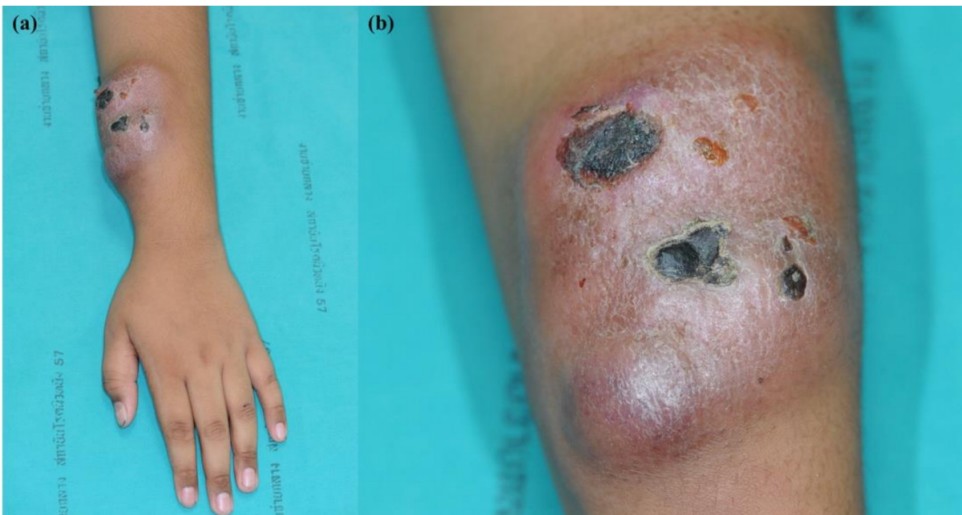

**Figure 1.** BPDCN (**a**) large solitary well-defined erythematous rubbery, fixed, non-tender mass on the left forearm, 8 cm in diameter; (**b**) ulceration and necrotic crusts.

His initial blood test results were Hb, 12.2 g/L (10.9–14.9 g/L); WBC, 8300/mm$^3$ (4500–14,500/mm$^3$); platelets, 319,000/mm$^3$ (150,000–400,000/mm$^3$); and lactate dehydrogenase, 208 U/L (140–290 U/L). No neoplastic cells were detected in the bone marrow or cerebrospinal fluid. Computed tomography (CT) demonstrated multiple subcentimeter lymph nodes in para-aortic, aortocaval, and mesenteric regions, along with moderate splenomegaly.

A skin biopsy revealed diffuse dermal infiltration by monomorphic agranular medium-sized blast cells sparing the grenz zone and epidermis (Figure 2). The first panel of immunostaining at the Institute of Dermatology showed positivity with TdT, CD79a, BCL2, and focal BCL6 (Figure 3) while CD3, CD5, CD10, CD20, and CD23 were negative. The initial diagnosis was lymphoma/leukemia cutis; the patient was then referred to Siriraj Hospital for a second opinion on the pathologic diagnosis and consultation for treatment. Repeat immunostaining for TdT, CD79a, and additional immunostaining showed positivity with CD123, TCL-1, CD4, CD56, CD99, CD45, focal TdT, and faint CD79a but PAX5, CD1a, CD3, CD5, CD7, CD8, CD10, CD14, CD15, CD20, CD23, CD33, CD34, CD68, CD117, and MPO were negative. The proliferation index by Ki-67 was 50–60%. (Figure 4) Thus, a final diagnosis of BPDCN was made.

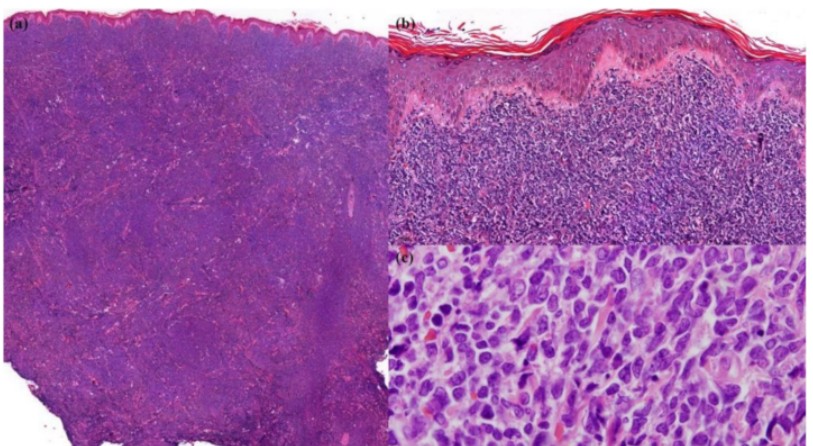

**Figure 2.** Skin biopsy. (**a**) Dense diffuse infiltration by neoplastic cells in the dermis and subcutaneous fat (Hematoxylin and Eosin (H&E), 20× original magnification); (**b**) sparing grenz zone and epidermis (H&E, 200× original magnification); (**c**) monomorphic agranular medium-sized blast cells (H&E, 1000× original magnification).

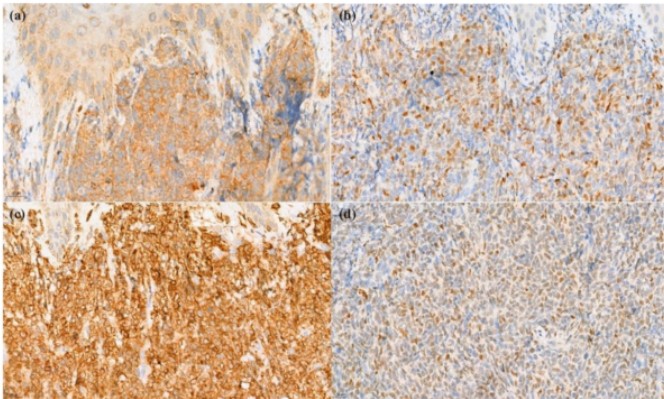

**Figure 3.** Immunohistochemical stain. (**a**) Positivity of CD79a (400× original magnification); (**b**) TdT (400× original magnification); (**c**) BCL2 (400× original magnification), and (**d**) BCL6 (400× original magnification).

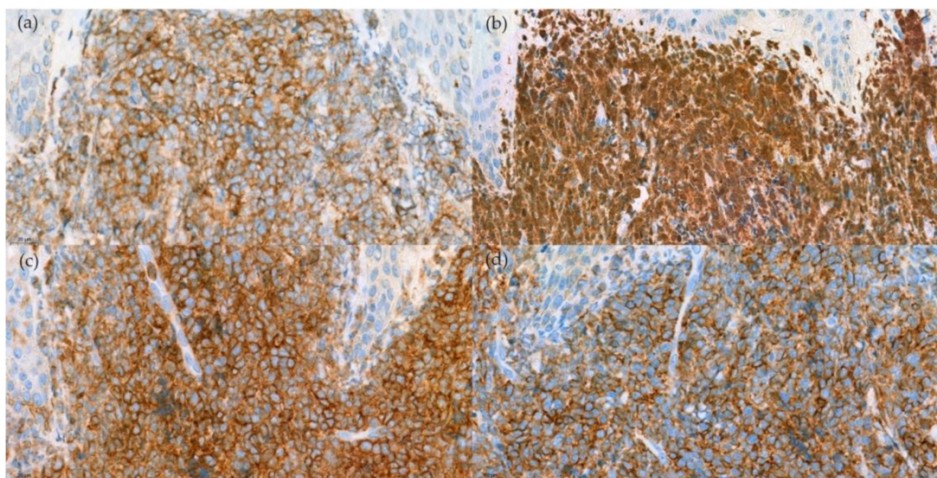

**Figure 4.** Immunohistochemical stain. (**a**) Positivity for CD123 (400× original magnification); (**b**) TCL-1 (400× original magnification); (**c**) CD4 (400× original magnification); and (d) CD56 (400× original magnification).

A positron emission tomography (PET)/CT scan was performed to identify the primary tumor and metastatic disease. Hypermetabolic thickening of the soft tissue was observed in the left forearm (DS4) and bilateral group 2 cervical lymph nodes (DS4), indicative of an active malignant process. Additionally, the left axillary lymph nodes showed a significant metabolic reaction (DS2). The bone scan was negative.

The patient was treated with chemotherapy modified from the AIEOP protocol for ALCL, comprising dexamethasone, methotrexate, ifosfamide/cyclophosphamide, cytarabine, VP-16, and doxorubicin. [5] No serious adverse events were observed. Based on the PET/CT scan results, the tumor was nearly completely eradicated following six courses of intensive chemotherapy. There was no matched donor for stem cell transplantation; hence, the patient received ALL maintenance therapy to achieve a long-term complete response. Upon follow-up, 15 months after diagnosis, no tumor recurrence was detected on PET/CT.

## 3. Discussion

Despite certain challenges in formulating the diagnosis, our case illustrated classic BPDCN in a child. In practice, BPDCN should be included in the differential diagnosis when skin lesions are suspected to be lymphoma/leukemia cutis. The results of the first immunostaining panel suggested the possibility of B lymphoblastic lymphoma/leukemia; however, the lack of CD10 expression was unusual because the B lymphoblasts generally express CD10. [6] The results of the second immunostaining panel rendered a definite diagnosis of BPDCN. A similar approach of first testing classic lymphocyte markers, followed by screening for precursor neoplasms, including BPDCN and myelomonocytic neoplasms, by using CD43 and CD56, was recently proposed for suspected BPDCN as shown in Figure 5. After excluding myelomonocytic neoplasm and cutaneous lymphoma, confirmation with PDC markers was required for BPDCN diagnosis. [7] More recently, a study suggested initial dual immunostaining for TCF4 and CD123 for any skin lesion suspected of BPDCN. If the tumor cells showed dual expression of TCF4 and CD123, further immunostaining was not required. [8] However, in a nationwide study from Japan, only 39 out of 2090 patients (1.9%) with primary cutaneous lymphoma were BPDCN. [9] Due to the very low incidence of BPDCN, the practicality of initial testing for dual expression of TCF4 and CD123 during work-up of lymphoma or leukemia cutis is indeterminate.

Skin biopsy suspicious of lymphoma/leukemia cutis

⬇

IHC panel for lymphoma and leukemia
CD3, CD10, CD20, CD33, CD34, CD68, CD79a, MPO, PAX5, TdT

| −

Screening panel of IHC for BPDCN
CD4, CD56

| +

IHC Markers for BPDCN
TCL1, CD123, CD303, TCF4

**Figure 5.** Immunohistochemistry approach for BPDCN [7,8].

Although there is no standard treatment for childhood BPDCN, the majority of studies advocate treating these individuals initially with an ALL-type chemotherapy regimen [3,10]. Our patient did not receive chemotherapy as part of the ALL protocol at the beginning because the clinical work-up did not reveal any blood or marrow involvement. Six courses of the modified AIEOP protocol were administered with good responses. Thereafter, maintenance phase chemotherapy for ALL was administered and the patient went into remission.

According to a recent review, BPDCN in children has a less aggressive clinical course and a more favorable prognosis. Treatment with high-risk ALL therapy regimens with central nervous system prophylaxis is recommended for pediatric patients. In children with relapsed, refractory, or high-risk disease at presentation, stem cell transplantation is recommended. New targeted therapies, including the recently FDA-approved tagraxofusp and anti-CD123 cytotoxin, show great promise for improved response rate [11].

In conclusion, we described a rare case of BPDCN in a child mimicking lymphoma/leukemia cutis. In contrast to adult patients, our patient had a favorable clinical course, similar to other pediatric cases. Moreover, our patient was treated with a regimen that differed from the established recommendation; thus, our case report suggests a successful alternative regimen for BPDCN in children.

**Author Contributions:** Conceptualization, V.V., P.S., and M.C.; writing—original draft preparation, P.P., T.K., C.T., and S.S.; writing—review and editing, S.S., V.V.; supervision, V.V., P.S., M.C., and S.S.; project administration, P.P. All authors have read and agreed to the published version of the manuscript.

**Funding:** This case report received no external funding.

**Institutional Review Board Statement:** Not applicable.

**Informed Consent Statement:** Written informed consent has been obtained from the patient to publish this paper.

**Acknowledgments:** The authors Chayamon Takpradit, Manasmon Chairatchaneeboon, and Sanya Sukpanichnant are supported by Chalermphrakiat Grant, Faculty of Medicine Siriraj Hospital, Mahidol University.

**Conflicts of Interest:** The authors declare no conflict of interest.

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
