# Peer review of "A Rare Case of Blastic Plasmacytoid Dendritic Cell Neoplasm in a Child Mimicking Lymphoma/Leukemia Cutis"

_dermatopathology, doi:10.3390/dermatopathology9040038_

Round 1

Reviewer 1 Report

The reviewer wishes to thank the editor and the authors for the opportunity to review this beautifully written, concise, well-illustrated manuscript.  The article describes a case of CD4 (+), CD56 (+), and CD123 (+) neoplasm consistent with BPDCN in a nine year old male that, despite the historically poor prognosis of BPDCN (usually described in the elderly), went into remission on a proposed alternate treatment regimen for BPDCN.

While this neoplasm appears to meet minimum criteria for the diagnosis of BPDCN, it may be helpful to the readers of this manuscript to further expand the discussion with the following:

1.       Diagnostic criteria for making the diagnosis of BPDCN (possibly with inclusion of TCL-1)

2.       Alternative differential diagnoses, possibly with a flow diagram for IHC so readers can easily follow the differences in staining patterns leading to this conclusion.

Additionally, if the concluding statement purposes this regimen as an effective alternative for BPDCN in children, additional discussion of differences in both regimens/ comparisons to each other are likely needed. 

Overall, a nicely written manuscript identifying a rare neoplasm in the pediatric population

Author Response

Point 1: Diagnostic criteria for making the diagnosis of BPDCN (possibly with inclusion of TCL-1)

Response 1: We affirm that the diagnostic criteria for BPDCN must include the TCL-1 marker, which must be positive for four out of five immunohistochemical markers; CD4, CD56, CD123, CD303, and TCL-1; in addition to the presence of cutaneous lesions with a blastic appearance of tumor cells.

As a result, we proceed to stain for TCL-1, and as anticipated, the result is positive (nuclear staining). We substitute the staining result image for the Ki-67 result in image 4 (b) so that the overall case report will be concise and contain only essential photographs.

In addition, the result of TCL-1 is included on line 93 of page 3.

Point 2: Alternative differential diagnoses, possibly with a flow diagram for IHC so readers can easily follow the differences in staining patterns leading to this conclusion.

Response 2: We provide a diagram representing our immunohistochemistry approach for BPDCN as an attachment and inserted on line 145 of page 5.

Point 3: Additionally, if the concluding statement purposes this regimen as an effective alternative for BPDCN in children, additional discussion of differences in both regimens/ comparisons to each other are likely needed. 

Response 3: According to the BPDCN, is an extremely rare hematologic malignancy in both adults and children. Consequently, there is currently no recommendation for a standard treatment regimen for BPDCN in children; although, the majority of reports treat pediatric patients through using ALL protocol. Howeve, m ost of these publications, meanwhile, did not mention the skin when treating BPDCN children with the ALL regimen. In addition, the report used the ALL protocol to demonstrate the unfavorable treatment outcome. As a result, our team has achieved a consensus on implementing the described treatment and maintenance of the the ALL regimen to prevent relapse.

We have inserted the statement "Although there is no standard treatment for childhood BPDCN, the majority of studies advocate treating these individuals initially with ALL-type chemotherapy regimens" on line 147 of page 5.

Lastly, I have conducted a spellcheck on the English language.

Sincere regards,

Reviewer 2 Report

This article is a report of a rare pediatric case of BPDCN. Epidemiology, clinical findings, immunohistochemical features, and treatment are also briefly summarized. There are only minor comments.

P3, lines 79-80: Describe the normal ranges.

P4, line 117: not clear what is meant by "nearly eradicated completely"; is it near or completely?

Author Response

Point 1: P3, lines 79-80: Describe the normal ranges.

Response 1: We insert the age-appropriate normal range of complete blood count for children as well as the LDH value on line 80-82 of page 3.

Point 2: P4, line 117: not clear what is meant by "nearly eradicated completely"; is it near or completely?

Response 2: Our intention is to clarify that the tumor has been almost completely eradicated (nearly eradicated) so that we have corrected the statement on page 5, line 119.

Sincere regards,
